# Biomechanical Analysis of the FlatFoot with Different 3D-Printed Insoles on the Lower Extremities

**DOI:** 10.3390/bioengineering9100563

**Published:** 2022-10-17

**Authors:** Chia-Yi Hsu, Chien-Shun Wang, Kuang-Wei Lin, Mu-Jung Chien, Shun-Hwa Wei, Chen-Sheng Chen

**Affiliations:** 1Department of Physical Therapy and Assistive Technology, National Yang Ming Chiao Tung University, Taipei 112, Taiwan; 2Department of Orthopaedics and Traumatology, Taipei Veterans General Hospital, Taipei 112, Taiwan; 3Department of Surgery, School of Medicine, National Yang Ming Chiao Tung University, Taipei 112, Taiwan

**Keywords:** 3D printing, insole, flat foot, range of motion, joint moment

## Abstract

Insoles play an important role in the conservative treatment of functional flat foot. The features of 3D-printed insoles are high customizability, low cost, and rapid prototyping. However, different designed insoles tend to have different effects. The study aimed to use 3D printing technology to fabricate three different kinds of designed insoles in order to compare the biomechanical effects on the lower extremities in flat foot participants. Ten participants with functional flat foot were recruited for this study. Data were recorded via a Vicon motion capture system and force plates during walking under four conditions: without insoles (shoe condition), with auto-scan insoles (scan condition), with total contact insoles (total condition), and with 5-mm wedge added total contact insoles (wedge condition). The navicular height, eversion and dorsiflexion angles of the ankle joint, eversion moment of the ankle joint, and adduction moment of the knee joint were analyzed, and comfort scales were recorded after finishing the analysis. Compared to the shoe condition, all three 3D printed insoles could increase the navicular height and ankle dorsiflexion angle and improve comfort. Among the three insoles, the wedge condition was the most efficient in navicular height support and increasing the ankle dorsiflexion angle.

## 1. Background

With a worldwide incidence between 10–25% [1,2], functional flat foot (FFF) is defined as temporary absent or abnormally low medial longitudinal arch height during weight bearing, with the capacity to regain the original foot shape when relaxed [3]. The classic presentation of FFF includes hindfoot valgus, midfoot abduction, and forefoot compensatory supination [4]. These deformities are proven to alter the kinematics and kinetics of the lower limbs [5], and are associated with increased odds of injuries [6]. Overstretching the plantar fascia will increase their tension and could lead to plantar fasciitis. FFF patients were shown to have thicker plantar fascia and significantly higher rates of plantar fasciitis [7]. Additionally, low arch height and over-pronated mid- and hindfoot-related tibia internal rotation will cause knee joint valgus [8,9], thus increasing the patellofemoral joint contact force [9], which may result in patellofemoral joint pain [10] and patella tendinopathy [11]. Meanwhile, increased knee valgus moment will cause higher joint contact force, accelerate knee joint lateral compartment cartilage wear, and increase the risk of knee joint valgus osteoarthritis [12,13,14].

Insoles play an important role in the conservative treatment of FFF. Through directly supporting the medial longitudinal arch and controlling the alignment of the heel, insoles can change the kinematics and kinetics of the knee joints [8,15,16,17] and theoretically reduce the risk of lower limb injury and degeneration. However, conventional customized insoles have the shortcomings of high labor and time costs. Additionally, since the biomechanical effects of insoles vary depending on their configuration [18], the therapeutic effect of conventional customized insoles might vary due to different ways of obtaining the foot model and diversity in technician practices. In recent years, three-dimensional (3D) printing proved to lower the costs of customized insole manufacturing [19]. The biomechanical effect of 3D printed insoles is reported to be comparable with that of conventional customized insoles [18]. Moreover, 3D printed insoles could be better integrated with the obtaining of computerized auto-scan foot models. Through standardized design and manufacturing, the therapeutic effects of 3D printed insoles could become more consistent and trustworthy. Currently, little information is available regarding the biomechanical effect of 3D printed insoles, especially on the lower extremities. Additionally, some physical therapists add wedges to customized insoles to correct gait during walking. As a result, there are some geometrical differences among auto-scan insoles, 3D printed insoles and 3D printed insoles with wedges. Those differences might affect individual gait. Therefore, the purpose of this study was to compare the differences in ankle and knee joint kinematics and kinetics and subjective comfort during level walking among three different 3D printed insoles that were fabricated with different ways of obtaining foot models and designs.

## 2. Methods

This prospective study was approved by the institutional review board of National Yang-Ming University. All participants provided informed consent before their participation in the study. Inclusion criteria included: (a) participants aged ≥20, (b) arch appeared during standing on tiptoe, (c) Foot Posture Index (FPI) [20] total score ≥6. FPI was evaluated in standing status with six items as follows: (1) talar head palpation, (2) curvature at the lateral malleoli, (3) inversion/eversion of the calcaneus, (4) talonavicular bulging, (5) congruence of the medical longitudinal arch, (6) abduction/adduction of the forefoot with respect to the rear foot. Each item was scored on a scale of −2, −1, 0, +1, +2 (0 for neutral, −2 for clear signs of supination, and +2 for clear signs of pronation), and all scores were summed. The final scores ranged from −12 to +12; a larger positive value indicates a more pronated foot. A previous study [20] found that the FPI is a useful tool to evaluate flat foot compared to radiographic analysis.

For the exclusion criteria, participants with recent insole or orthosis usage (1 month), operative history of foot and ankle fracture/reconstruction/osteotomies, and medical history associated with connective tissues were excluded. Ten participants (five males and five females; age: 30.7 ± 12.86 years; height: 165.7 ± 9.07 cm; weight: 63.67 ± 9.71 kg) were recruited for this study. The average FPI score was 7.13 ± 1.46 (left side) and 7.38 ± 2.07 (right side).

### 2.1. 3D-Printed Insole Fabrication

Three types of 3D printed insoles, consisting of an auto-scan insole, a total contact insole and a medial wedge insole, were designed using software and equipment developed by Joy Sunrise Research Inc., Taipei, Taiwan. The bases of auto-scan insoles and total contact insoles were fabricated with a ¾ foot length, and the bases of medial wage insoles were fabricated at full foot length. Insole bases were constructed using Fused Deposition Modeling (FDM) 3D printing technology with thermoplastic polyurethane (TPU). The diameter of the TPU filament was 1.75 mm with a printing thickness of 0.2 mm. The nozzle temperature was set at 250 °C. The printing velocity was 30 mm/s. The 3D printer (Joy Sunrise Research Inc., Taipei, Taiwan) was developed especially for this study. The time required for printing each TPU insole base was approximately 1.5 h. After the fabrication of the insole bases, an ethylene-vinyl acetate copolymer (EVA) top was attached to complete the insole. The reason for choosing TPU for the insole base was because it had enough hardness to support the foot arch. Conversely, the soft EVA top was chosen to absorb impact energy during walking. The manufacturing procedures for the three types of insole are described below:Auto-scan insoles: the participants were asked to stand on a pressure sensor with body weight equally distributed across both feet using an auto-scan system (Joy Sunrise Research Inc., Taipei, Taiwan) (Figure 1). Subsequently, the software automatically determined the module of the insole based on the distribution of pressure.Total contact insoles: the geometric shape of the foot was obtained by scanning a foam box stepped in by the participants via a 3D scanner (iSense 3D, 3D system Inc., Washington, DC, USA) (Figure 2).Medial wedge insoles: 5-mm wedges were added through the medial side of the foot to the abovementioned scanning results using foot model editing software (InsoleModuleDesign, Joy Sunrise Research Inc., Taipei, Taiwan) (Figure 3).

### 2.2. Experimental Procedures

Gait and motion analysis data were collected in the human Motion Analysis Laboratory at National Yang Ming Chiao Tung University. An eight-camera 3D Vicon (MX T20, Vicon Motion Systems Ltd., Oxford, UK) motion analysis system sampling at 100 Hz and an AMTI force plate (Advanced Mechanical Technology Inc., Watertown, MA, USA) sampling at 1000 Hz were used to collect kinematic and kinetic data. Reflective markers adapted from a previous study [21] were placed bilaterally on the anterior and posterior superior iliac spine, greater trochanter, lateral and medial femoral epicondyles, head of the fibula, tibial tuberosity, and lateral and medial malleolus. To measure foot motion inside a standard shoe, the shoe was prepared with four additional cut-outs for placing markers at the heel, navicular tuberosity, and tuberosity of the second and fifth metatarsals (Figure 4). The markers were attached on the dominant leg. Each participant’s dominant leg was determined through a ball-kicking test, and all participants were right-side dominant.

Prior to data collection, participants were given a 5-min practice trial to familiarize themselves with the experimental surroundings (Figure 5). In addition, a static trial was performed to determine the joint center and the neutral joint orientations before each experimental condition. Participants were asked to perform five trials of level walking at a self-selected speed under four conditions: (1) in standard shoes without insoles (shoe condition), (2) in shoes with auto-scan insoles (scan condition), (3) in shoes with total contact insoles (total condition), and (4) in shoes with medial wedge insoles (wedge condition). A 5-min break was provided between conditions. The order of the experimental conditions was randomized across participants. After finishing the dynamic experiments, participants were asked to report their wearing comfort scale in a range from 0–10 (most uncomfortable to most comfortable).

All kinematic and kinetic human motion data were processed using Matlab (MathWorks Inc., Natick, MA, USA). Ankle and knee joint moments were calculated using the inverse dynamics method [21]. The calculated joint moments were normalized by multiplying the body weight and leg length. The maximum and minimum moment values at characteristic peaks during the stance phase were obtained from each participant’s average curve across the five trials.

### 2.3. Statistical Analyses

Statistical analyses were performed using IBM SPSS statistics software, version 24.0 (IBM, Armonk, NY, USA). The Friedman test was performed to verify if there were significant differences among the four conditions. Post-hoc analysis with the Wilcoxon test was carried out when significant differences were found. The statistical significance level was set at 0.05.

## 3. Results

### 3.1. Navicular Height

Navicular height was significantly higher under the three conditions with insoles, increasing by 40%, 53%, and 59% in the scan, total, and wedge conditions, respectively (Table 1).

### 3.2. Ankle Joint Peak Eversion Angle and Frontal Plane Motion

The peak value of the ankle joint eversion angle decreased in the total and scan conditions by 13% and 10%, respectively, as compared with the shoe condition. However, the peak eversion angle under the wedge condition was elevated by 33% compared with the shoe condition. The differences in peak ankle eversion angle between conditions did not reach statistical significance (*p* = 0.293) (Table 1).

During the whole early stance phase (20~80%), the ankle joint eversion angle under the total condition was lower compared to under the shoe condition and similar between the scan and total conditions, but significantly higher in the wedge condition (Figure 6).

### 3.3. Ankle Joint Peak Dorsiflexion Angle and Sagittal Plane Motion

The peak ankle dorsiflexion angle was significantly higher in the scan and wedge conditions, with increases of 21% and 31%, respectively, compared to the shoe condition. Although there was also higher peak ankle dorsiflexion observed in the total condition, the difference did not reach statistical significance (Table 1).

The increase in ankle dorsiflexion angle occurred mainly in the late stance phase and reached statistical significance. Among all conditions, the wedge condition had the largest increase (Figure 7).

### 3.4. Ankle Joint Frontal Plane Moment

Compared with the shoe condition, the peak ankle eversion moment was lower in the scan and total conditions, by 6% and 9% respectively, and higher by 1% in the wedge condition. However, these differences did not reach statistical significance (*p* = 0.668).

Compared with the shoe condition, the total condition had lower ankle joint eversion moment. Moreover, in the mid-stance phase, the scan condition revealed similar eversion moment, and the wedge condition had significantly higher eversion moment (Figure 8).

### 3.5. Knee Joint Frontal Plane Moment

Compared with the shoe condition, the peak adduction moment of the knee joint increased in the total and wedge conditions by 5% and 9%, respectively, and increased in the scan condition by 3%. However, these differences were not significant (*p* = 0.516).

In the gait cycle, the scan and total conditions revealed similar adduction moments compared with the shoe condition; however, in the pre-stance phase, there was a slight increase in the adduction moment in the wedge condition (10~30%). These differences did not reach statistical significance (Figure 9).

### 3.6. Comfort Scale

The comfort scale values were 5.00 ± 0.81 (shoe), 6.40 ± 2.59 (scan), 8.45 ± 1.01 (total), and 7.90 ± 1.45 (wedge), respectively. There were significant differences between the total and wedge conditions and the shoe condition, and between the total and scan conditions (Figure 10).

## 4. Discussion

The results of this study revealed that 3D printed insoles could effectively improve ankle joint kinematics and kinetics, as well as subjective comfort, during level walking. The static and dynamic navicular height, ankle dorsiflexion angle, and comfort score all significantly improved among the three different 3D printed insoles, and the difference in ankle dorsiflexion occurred mainly in the late-stance phase. Although there were no significant differences in peak value ankle eversion angle and moment when using 3D printed insoles, total contact insoles decreased the ankle eversion angle and moment in the mid-stance phase (20–80%). As for knee frontal plane moment, no significant differences were observed in the conditions with or without 3D printed insoles.

Due to a loss of medial arch support, the typical FFF deformity includes a more vertically ordinated talus in relation to the tibia, and the ankle joint therefore sits in a relatively plantarflexed position [22,23]. Meanwhile, compensatory forefoot supination might further deteriorate ankle plantarflexion to ensure that the forefoot fully contacts the ground in the late-stance phase. In our study, all 3D printed insoles could effectively correct ankle plantarflexion in the late-stance phase. Among them, insoles that fit the foot’s geometric shape with additional 5-mm medial wedges (wedge condition) had the best correction effect, and the difference mainly occurred in the late-stance phase. Based on this result, our explanation was that the possible corrective effect might result from medial support, especially on the forefoot. Peng et al. reported reduced patellofemoral contact force, peak ankle contact force, and ankle eversion angle and moment with the usage of medial arch and forefoot posting insoles [8]. Although the effect of forefoot medial support might differ due to the differences in inclusion criteria, experimental procedure, and the hardness of support materials, our study further proved the benefits of forefoot medial support.

Previous studies showed contradictory results regarding the ankle eversion effect of insole usage. One previous study reported that insoles improved the eversion angle, except for initial eversion during the heel strike [24]. However, other studies revealed that insoles might not control ankle eversion [15,25]. Although our results revealed that insoles that fit the foot’s geometric shape might have some positive effect on ankle eversion control in the mid-stance phase during gait, our three insoles did not improve peak ankle eversion angle and moment. Although Kido et al. reported improved static ankle eversion after the usage of medial-wedged therapeutic insoles [26], the wedge condition had the largest ankle eversion peak angle and moment in our results. As a result, insoles without wedges could not obtain a better correction effect because of an insignificant increase in ankle eversion moment. Since peak ankle eversion happened in the early-stance phase [27,28], there was a possible explanation: the hardness of our insoles was not strong enough to control the hindfoot during the heel strike. However, the 3D printed insoles exhibited a porous structure. As a result, we could fabricate 3D printed insoles in future study with a higher printed density to increase hardness and thus increase the ability to control the hindfoot during walking. Regardless, the 3D printed insole consisted of a hard TPU base (supporting foot arch) and soft EVA cover (absorbing impact) and was effective at raising navicular height and subject comfort.

In order to obtain 3D foot models, we employed auto-scan insoles which utilized foot pressure distribution to determine the shape of the insoles. The results proved that scan insoles could effectively increase navicular height, increase ankle dorsiflexion angle, and provide comfort compared to conditions without insoles (shoe condition). However, decreased navicular support and comfort were noted when compared with foot geometry-generated insoles (total condition). Although auto-scan insoles could provide more consistent therapeutic effects and might become mainstream in insole manufacturing, there is still room for improvement in the conversion from foot pressure to the insole module. Meanwhile, other parameters such as deformity correction could be added in software editing.

Regarding the effect of fabricating insoles via 3D printing, the results indicated that one feature of 3D printing an insole is reduced manufacturing time and material cost compared to a traditional customized insole. In manufacturing time, a traditional customized insole takes approximately one week to fabricate because it needs more labor force. By comparison, a 3D-printed insole could be fabricated within 2 h (1.5 h in printing time and 0.5 h in editing the insole model). Additionally, the materials for 3D printed insoles cost approximately 22 USD including the TPU base and EVA top. The sale price of a traditional customized insole in Taiwan is approximately 250 USD; by comparison, 3D printed insoles could be sold well under 250 USD. Since they are automatically fabricated by the 3D printer, they may reduce the problems with handmade insoles as well, especially in adjusting foot arch height. Additionally, increasing numbers of 3D printers could result in the mass production of customized insoles.

This study had limitations that should be noted. (1) The study was designed to evaluate the immediate biomechanical effect; therefore, we did not evaluate the long-term impact of our insoles. (2) When evaluating the accuracy of printed insoles, we compared the maximum width of a real insole and a digital insole. There was only a 0.5% difference between the digital file (82.37 mm) and the actual fabricated insole (82.81 mm). This difference could be because of thermal expansion of the TPU material. (3) We used low-topped flat shoes in all experimental conditions. Therefore, the effect may differ when used with daily shoes. (4) The severity of our participants’ conditions was not consistent. The effect of insoles may vary across deformity levels. (5) There were only ten participants included in our study. Further studies should include a larger population to verify the results. Furthermore, different walking conditions should be included in future studies.

## 5. Conclusions

We compared the biomechanical effect of three different 3D printed insoles fabricated from TPU bases and EVA covers on the lower ankle and knee joints. All three types of 3D printed insoles (scan, total and wedge) could provide significant arch support, increase ankle dorsiflexion angle, and improve comfort. Since relatively better results were obtained in the wedge condition, deformity-specific adjustment should be included in insole configuration.

## Figures and Tables

**Figure 1 bioengineering-09-00563-f001:**
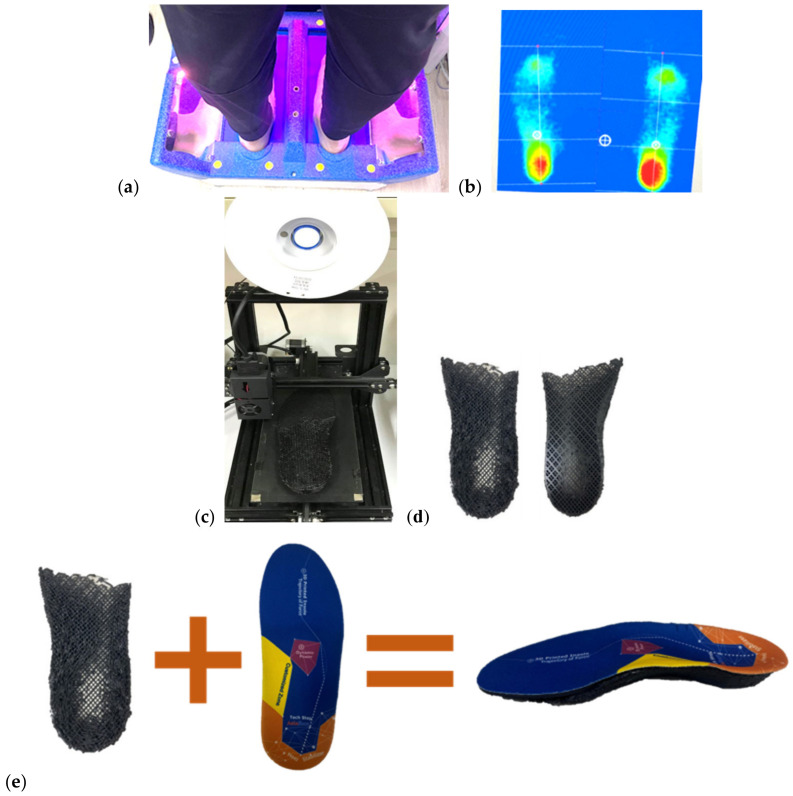
Procedure for fabricating an auto-scan insole: (**a**) Participant stands on the foot pressure system; (**b**) Foot pressure is measured; (**c**) Pressure file is sent directly to the 3D printer; (**d**) Insole base is printed automatically; (**e**) EVA top is glued to TPE insole base.

**Figure 2 bioengineering-09-00563-f002:**
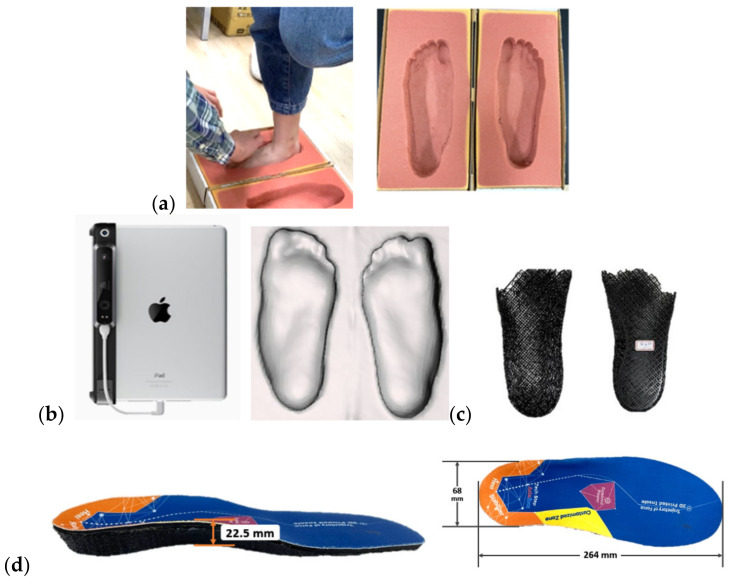
Procedure for fabricating a total contact insole: (**a**) Participant stands on a foam box to acquire geometry of the foot; (**b**) A 3D scanner is used to digitize the real foot from foam box into a computerized foot model; (**c**) Insole base with ¾ foot length is printed; (**d**) EVA top is glued to TPE insole base. Insole dimensions from one participant shown.

**Figure 3 bioengineering-09-00563-f003:**
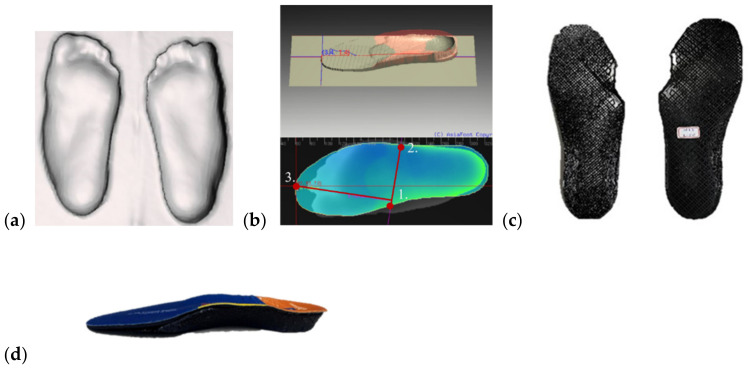
Procedure for fabricating a medial wedge insole: (**a**) Foot model is acquired from foam box as in Figure 2; (**b**) Foot model is edited to add medial wedge; (**c**) Insole base with full length is printed; (**d**) EVA top is glued to TPE insole base.

**Figure 4 bioengineering-09-00563-f004:**
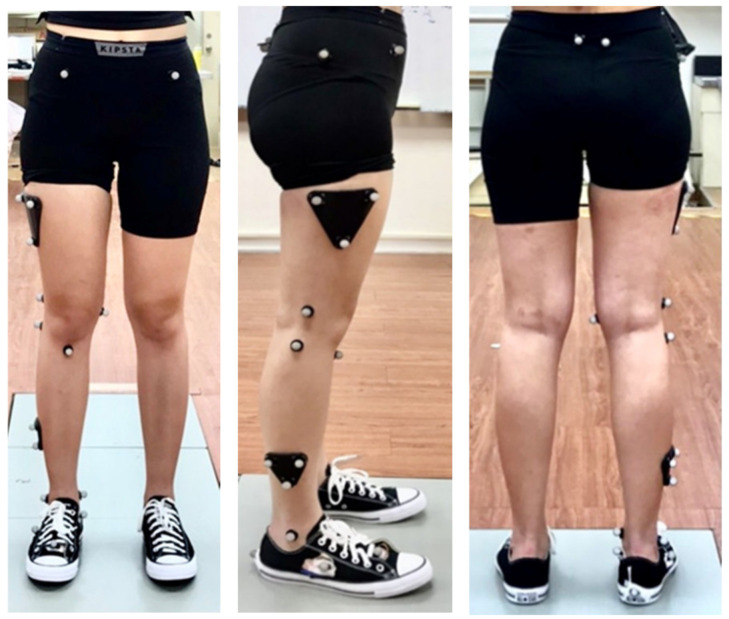
Locations of the reflective markers.

**Figure 5 bioengineering-09-00563-f005:**
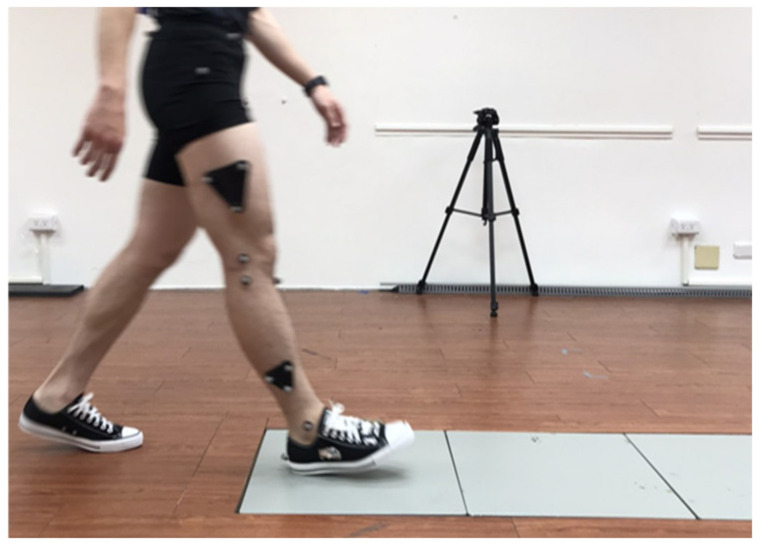
Experimental process.

**Figure 6 bioengineering-09-00563-f006:**
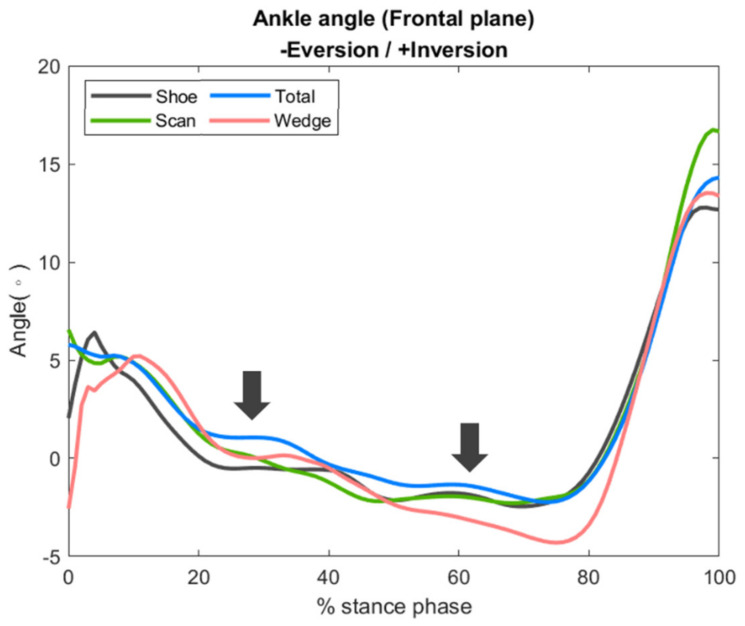
Frontal plane kinematics of the ankle joint, with arrows indicating the time points of greatest difference.

**Figure 7 bioengineering-09-00563-f007:**
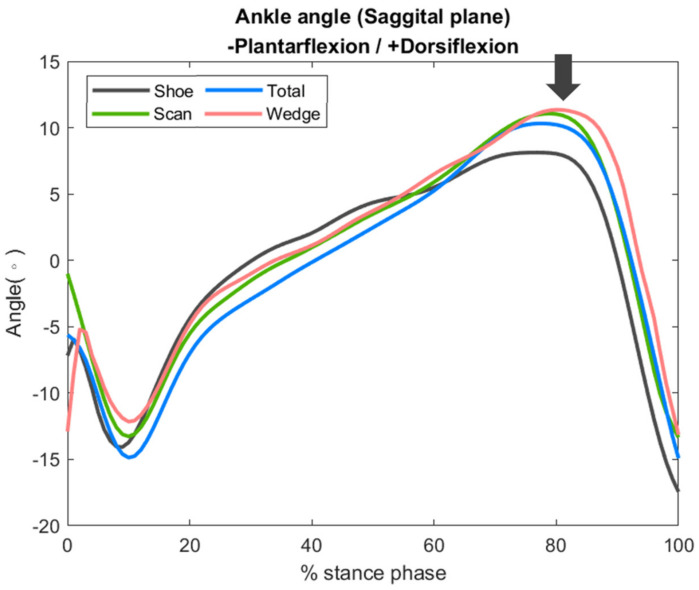
Sagittal plane kinematics of the ankle joint, with the arrow indicating the time point of the greatest difference.

**Figure 8 bioengineering-09-00563-f008:**
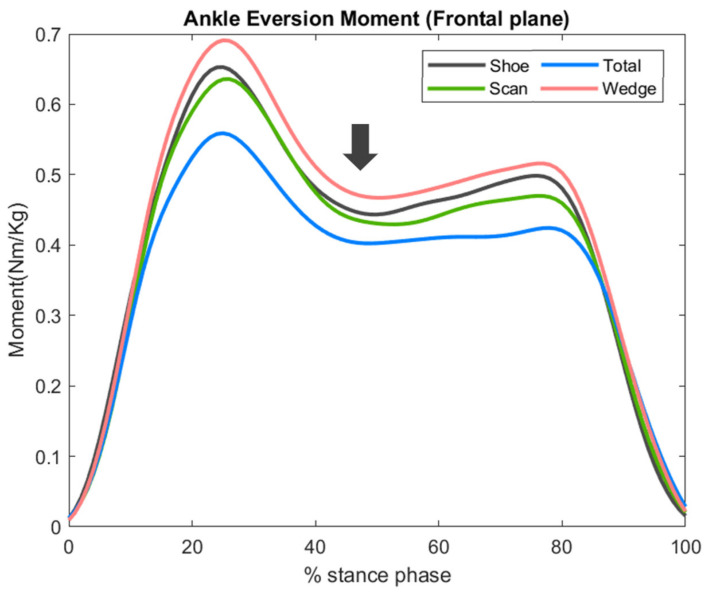
Sagittal plane kinetics of the ankle joint, with an arrow indicating the time point of the greatest difference.

**Figure 9 bioengineering-09-00563-f009:**
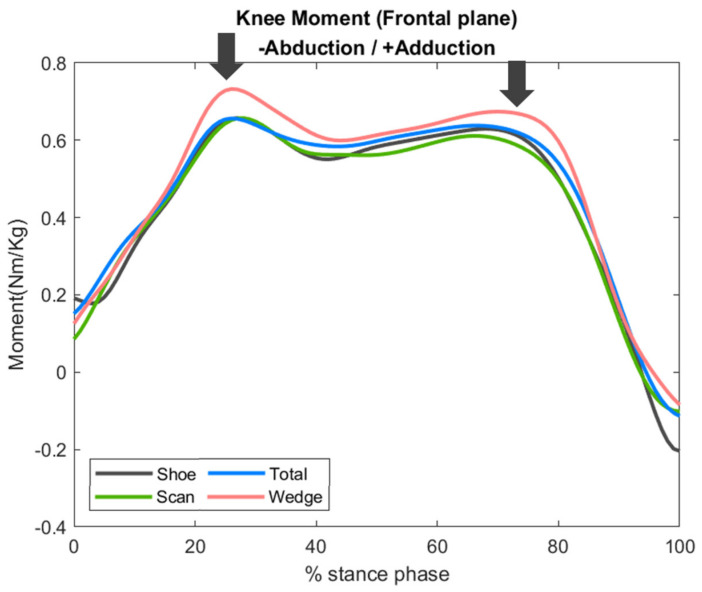
Frontal plane kinetics of the knee joint, with arrows indicating the time points of greatest difference.

**Figure 10 bioengineering-09-00563-f010:**
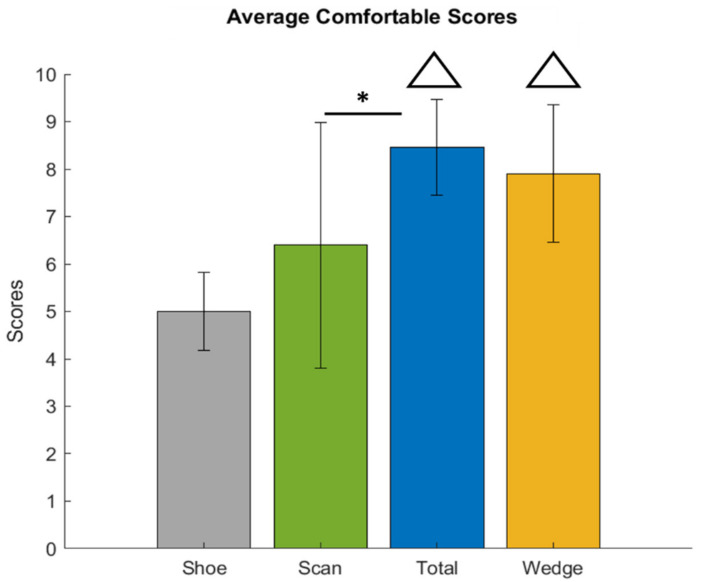
Average comfort scores. △: compared with the Shoe condition (*p* < 0.05); *: compared with Scan condition (*p* < 0.05).

**Table 1 bioengineering-09-00563-t001:** Average lower limb kinematic and kinetic values.

	Shoe	Scan	Total	Wedge
Lowest navicular height in static standing (mm)	20.1 (11.6)	29.4 (13.0) *	30.4 (11.3) *	32.2 (12.4) *
Lowest navicular height in dynamic walking (mm)	19.8 (12.0)	27.9 (12.0) *	30.3 (11.7) *	31.6 (11.6) *
Ankle eversion angle peak value (°)	4.7 (6.0)	4.0 (7.4)	4.2 (6.4)	6.2 (6.5)
Ankle eversion moment peak value (Nm/kg)	0.67 (0.44)	0.65 (0.39)	0.63 (0.44)	0.70 (0.27)
Ankle dorsiflexion angle peak value (°)	9.6 (3.4)	11.6 (3.2) *	11.0 (3.0)	12.5 (4.3) *
Knee adduction peak value (Nm/kg)	0.77 (0.38)	0.72 (0.36)	0.81 (0.43)	0.79 (0.24)

Note: Data are expressed as mean (standard deviation); * *p* < 0.05 (compared to Shoe group).

## Data Availability

The data presented in this study are available on request from the corresponding author.

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
