# Peer review of "Biomechanical Analysis of the FlatFoot with Different 3D-Printed Insoles on the Lower Extremities"

_bioengineering, 2022, doi:10.3390/bioengineering9100563_

Round 1
Reviewer 1 Report
1. It is recommended to describe each sub-figures in detail in the caption to Figure 1.
2. It is recommended to improve the quality of Figure 2. It is recommended to show a more detailed image of 3d printed insoles. Give the overall dimensions of the samples.
3. It is recommended in section 2.1. D-printed insole fabricating to add the insoles printing time. How was the post-processing of products carried out after printing? Add weight of printed insoles. Explain why polyurethane (TPU) was chosen for the manufacture of products.
4. It is recommended to discuss how the results obtained can be applied in the engineering practice of designing and manufacturing insoles. How cost-effective 3d printing can be in mass production of insoles?
Author Response
"Please see the attachment."

Reviewer 2 Report
The authors studied the use of 3D printing for customised insole designs to improve the biomechanical effects on the lower extremities in flatfoot participants. While the manuscript is generally well executed, there are several issues that should be addressed before further consideration for publication.
1. Figure 1 does not provide much information on its own. Suggest to edit the Figure into a flow chart with key information on how the scans are done and what kind of information/outputs are obtained.
2. Similarly, for Figure 2, the images do not provide much information. In particular, Figure 2(e) shows a 3D printer, but no information is provided on the 3D printing set up or process. Suggest the authors to also describe the 3D printing or additive manufacturing process using ISO/ASTM terminology.
- Wickramasinghe et al. (2022), Flexural behavior of 3D printed bio-inspired interlocking suture structures, Materials Science in Additive Manufacturing 1 (2), 9
- Teng et al. (2022), A methodology to design and fabricate a smart brace using low-cost additive manufacturing, Virtual and Physical Prototyping 17 (4), 932-947
3. Any discussion on the quality of scan and fabrication. Are there any deviations between the digital file and actual fabricated insoles? How will these affect the results?
Author Response
"Please see the attachment."

Reviewer 3 Report
The manuscript was designed to compare the differences in the ankle and knee joint kinematics, kinetics, and subjective comfort during walking using three printed insoles.
The introduction is concise and clearly states the rationale of the study. I have a concern regarding the use of three insole types. There is no rationale for comparing them. I assume that a couple of sentences in the intro may resolve this issue.
In the methods, I recommend the authors to explain some details of the tests a little further. For instance, what does the FPI test refer to, and why a cutoff of 6 was chosen? In addition, what are the implications of having a side asymmetry in the FPI?
Please, replace height with stature and weight with body mass or change the latter's unit.
I am concerned with the fact that only one side was corrected. Why was the test performed in only one segment while both were diagnosed with FFF? Isn't there an effect regarding fixing only one side? Is there a possible compensatory effect?
What a “standard shoe” refers to? It requires more detailed information.
Is the average of 5 trials representative? Please, expand the arguments to justify why a reduced number of trials were performed.
It is important to emphasize that most adaptations will likely occur after the participants get used to the insoles. I doubt they would present large acute differences. Please, inform how long the participants had the opportunity to familiarize themselves with the insoles.
Figures are difficult to visualize with no standard deviation information. Please, consider using a color figure to provide a better idea to the reader (and the reviewer as well).
I would suggest more caution in the discussion, as not all conditions have changed the data in the same direction. For instance, note figure 7, in which the “wedge” insole showed a greater ankle eversion moment in the frontal plane. Further, not all conditions experienced a significant change.
Figure 9 requires revision in the information of the p-value, as it is not referring to “each” condition but the indicated conditions.
Please check the word “hard-ness” in line 221.
Author Response
"Please see the attachment."

Round 2
Reviewer 2 Report
NA